# Towards Multi-Objective Object Push-Grasp Policy Based on Maximum Entropy Deep Reinforcement Learning under Sparse Rewards

**DOI:** 10.3390/e26050416

**Published:** 2024-05-12

**Authors:** Tengteng Zhang, Hongwei Mo

**Affiliations:** College of Intelligent Systems Science and Engineering, Harbin Engineering University, Harbin 150001, China; zttdouble@hrbeu.edu.cn

**Keywords:** maximum entropy deep reinforcement learning, full convolutional network, sparse rewards, grasping decision-making

## Abstract

In unstructured environments, robots need to deal with a wide variety of objects with diverse shapes, and often, the instances of these objects are unknown. Traditional methods rely on training with large-scale labeled data, but in environments with continuous and high-dimensional state spaces, the data become sparse, leading to weak generalization ability of the trained models when transferred to real-world applications. To address this challenge, we present an innovative maximum entropy Deep Q-Network (ME-DQN), which leverages an attention mechanism. The framework solves complex and sparse reward tasks through probabilistic reasoning while eliminating the trouble of adjusting hyper-parameters. This approach aims to merge the robust feature extraction capabilities of Fully Convolutional Networks (FCNs) with the efficient feature selection of the attention mechanism across diverse task scenarios. By integrating an advantage function with the reasoning and decision-making of deep reinforcement learning, ME-DQN propels the frontier of robotic grasping and expands the boundaries of intelligent perception and grasping decision-making in unstructured environments. Our simulations demonstrate a remarkable grasping success rate of 91.6%, while maintaining excellent generalization performance in the real world.

## 1. Introduction

Most traditional robotic grasping techniques heavily depend on object labels [1,2] and are data-driven [3,4]. However, when confronted with unknown objects in unstructured and complex environments, the capabilities of autonomous learning, active adaptation, and generalization become essential for achieving skillful manipulation. The scenarios of robotic grasping in everyday life are wide-ranging, covering both single-target and multi-target grasping. Among these, the greatest challenge lies in successfully grasping unstructured and unknown objects. The progressive development of computer vision technology has paved the way for significant advancements in robotic manipulation skills, particularly through the application of deep reinforcement learning methods [5]. These advancements have laid a strong theoretical foundation for intelligent robotic manipulation in various complex tasks.

Nonprehensile manipulation refers to the interaction between a robot and an object without a specific grasping task. This form of manipulation encompasses a range of actions, including pushing, poking, hitting, hooking, rotating, flipping, throwing, squeezing, and twisting. Robotic manipulation can be classified into two categories: stabilizing the object through grasping and performing unconstrained manipulation when grasping is not possible. However, nonprehensile manipulation not only involves the relationship between the manipulator, the object, or the tool but also requires complex dynamic models of the object and the environment [6,7]. This entails developing intricate mathematical models to capture dynamic factors such as sliding friction, gravity, inertia, and motion planning for moving objects. Additionally, utilizing known object shapes, poses, materials, and desired trajectories for computation purposes can be expensive and challenging to adapt to new objects and environments [8].

Ideal robotic grasp technology must meet certain requirements. Firstly, flexibility is necessary for model-free unknown objects. Secondly, it must ensure high reliability in selecting objects from dense, cluttered, or obstacle-rich environments. Thus, this paper proposes a maximum entropy deep reinforcement learning for dexterous grasping, which combines fully convolutional networks (FCNs) and attention mechanisms to achieve higher feature extraction efficiency in different task scenarios. The key contributions can be outlined as follows:(1)Design a maximum entropy deep reinforcement learning grasping method based on an attention mechanism to address complex and sparse reward tasks while eliminating the trouble of adjusting hyper-parameters in unstructured grasping environments.(2)Design an experience replay mechanism to reduce data correlation and combine advantage functions to enhance reasoning and decision-making abilities in complex environments.(3)Design object affordance perception based on space-channel attention to make robots more flexible in dealing with various complex grasping tasks.(4)Our proposed method has generalization ability from simulation to real world. For cluttered situations, the experimental results indicate the grasping rate of unknown objects is up to 100% and 91.6% for single-object and multi-object, respectively.

The remainder of this paper is organized as follows. Section 2 briefly presents the preliminaries and problem formulation. Section 3 introduces push-grasp policy design, and Section 4 presents the experimental results and learning process. Finally, Section 5 concludes this work.

## 2. Related Work

Previous research primarily focused on geometric variations such as object grasp position and shape [9,10]. Zeng et al. [11] used Q-learning to choose discrete actions in a pixelwise manner and map the pixel coordinates to a real-world location. However, sparse rewards made it difficult to find a reward signal while performing a grasping operation; thus, it did not learn how to execute a given task. The objects were often pushed out of the workspace, and even when it was not necessary, pushing actions were taken, leading to a series of grasping and pushing actions. In [12], the pushing action was executed only when no object is graspable judging by a grasp detect algorithm. The robot only focused on grasping objects that were aligned with the bin wall or boundary, resulting in poor success rates. Separately, in order to grasp the objects placed in well-organized shapes, Chen et al. [13] employed a Deep Q-Network (DQN) to guide the robot in actively exploring the environment of the objects placed around highly randomly until a suitable grasp affordance map was generated. This data-driven deep reinforcement learning method results in improper selection of many grasping points due to insufficient training cases, with time-consuming training iterations and low grasping efficiency and success rates. Generally, the manipulators cannot recognize objects accurately in cluster scenes from a single viewpoint and cannot make the environment better for grasping.

Gang et al. [14] combined the pushing and grasping actions by an improved deep Q-network algorithm with an RGB-D camera to obtain the information of objects’ RGB images and point clouds from two viewpoints, which solved the problem of lack of information missing. To reduce the complexity of strategy learning, Chen et al. [15] made use of the twin delayed deep deterministic policy gradient to train policy that determines where to start pushing and pushing direction according to current image. They proposed a framework for robots to pick up the cluttered objects based on deep reinforcement learning and a rule-based method. Similar to [14,15], a double experience replay was set up to increase the search to learn efficient push and grasp policy in a tote box. However, only depth image was considered in their work, and so the test results for novel unknown objects was not perfect. More recent research makes it possible to train robot to learn synergies between pushing and grasping in dense clutter [16,17,18,19]. These methods utilize visual observations for end-to-end decision-making without using object-specific knowledge. Their test scenarios in the randomly cluttered challenge did not indicate the level of clutter, and the push performance was not evaluated with the arranged object challenge.

Although Lu et al. [19] proposed an attention module that includes target saliency detection and density-based occlusion area inference, the sparse reward leads to low robot motion efficiency, and inefficient pushing exploration actions also impact the success rate. Effectively grasping objects in a cluttered environment can be achieved through a novel approach that combines prehensile and non-prehensile manipulation policies. Kalashnikov et al. [20] introduce a scalable vision-based reinforcement learning framework named QT-Opt, which enables robots to learn how to pick up objects and execute non-prehensile pre-grasp actions. Kiatos et al. [18] designed an experiment to learn a direct correlation between visual observations and actions, and it is trained in a comprehensive end-to-end manner. Without assuming a segmentation of the scene, the grasping policy accomplishes robust power grasps in cluttered environments. Yuan et al. [21] trained policy end-to-end using a CNN-based deep Q-learning algorithm that maps raw pixels to state-action values, which are then transferred to the real world with supervised examples. Arneqvist et al. [22] emphasized the issue of transferring knowledge within a similar family. To address this, the variational policy embedding learning for adaptive master policy across similar Markov Decision Processes (MDPs) was proposed. Thus, this enables policy transfer even without pre-trained datasets. Meanwhile, the CNNs based on Monte Carlo tree search were used to train cup placement strategies [23]. The aim is to optimize enhanced strategies for simulation-to-real transfer and achieve domain-agnostic policy learning.

More closely related to our work is that of Zeng et al. [11]. Our method combines the depth information of objects with reinforcement learning to obtain adaptive strategy to enable a robot to learn pushes actively and purposefully and achieve better grasps. The grasping skills for novel objects have been well generalized in the real world. Compared with the previous works, the proposed method has stronger consistency and robustness. Learning expressive energy policy from Soft Q-Learning and combining non-strategic updates with Soft Actor-Critic is conducted to maximize expected returns and entropy in random situations. The prioritized experience replay is meant to reduce data correlation, and the advantage function improves the reasoning and decision-making ability of deep reinforcement learning in complex manipulation tasks. Finally, it is important to break through the possibility boundaries of autonomous intelligent perception and operations in unstructured environments.

## 3. Preliminaries and Problem Formulation

### 3.1. Model Description

Deep learning, a branch of machine learning, typically involves multiple layers of nonlinear operational units that utilize the output of the previous layer as input, automatically extracting deep feature from vast amounts of training data. It has achieved significant success in areas such as image processing, speech recognition, natural language processing, and robot control. Compared to traditional multilayer neural network algorithms, deep learning effectively mitigates gradient dispersion and local optima, alleviating the curse of dimensionality associated with high-dimensional data. Representative structures of deep learning include deep belief networks, stacked autoencoders, recurrent neural networks, and convolutional neural networks (CNNs) [24,25]. Reinforcement learning enables agents or robots to learn decision-making through millions of interactions across diverse domains and environments. Therefore, integrating the perceptual capabilities of deep learning with the decision-making abilities of reinforcement learning represents an intelligent approach that more closely resembles human thinking, achieving direct control from raw input to output through end-to-end learning. Especially in unstructured and complex scenarios, deep reinforcement learning plays a pivotal role in enhancing the efficiency, success rate, and robustness of robot grasping.

The process of deep reinforcement learning can be defined as follows: an agent interacts with environment, collecting experiences in the form of state-action-reward sequences. These experiences are then used to train a deep neural network, which learns to approximate either a value function or a policy function. The value function estimates the expected future reward for a given state or state-action pair, while the policy function directly maps states to actions. Through iterative optimization, the agent continuously improves decision-making strategy, aiming to maximize the cumulative reward over time. This end-to-end learning process allows the agent to directly learn control strategies from raw input data, enabling it to adapt to complex and unstructured environments with high efficiency, success rate, and robustness. Deep reinforcement learning algorithms can be categorized into three types: value-based reinforcement learning, policy-based reinforcement learning, and model-based reinforcement learning [26]. DQN improves upon traditional learning methods based on experience replay mechanisms, primarily in three aspects: (1) approximating the value function using a deep CNN; (2) reducing data correlation during training; and (3) independently establishing a target network to handle TD errors (temporal difference errors).
(1)$$L(\theta )=E_{s,a\sim \rho (\cdot )}[{\left(TargetQ-Q(s,a;\theta )\right)}^{2}]$$
(2)$$TargetQ=E_{s^{\prime }\sim S}[r+\gamma \max_{a^{\prime }}Q(s^{\prime },a^{\prime };\theta^{\prime })|s,a]$$
(3)$$\nabla_{\theta }L(\theta )=E_{s,a\sim \rho (\cdot );s^{\prime }\sim S}[\theta_{t}+\alpha (r+\gamma \max_{a^{\prime }}Q(s^{\prime },a^{\prime };\theta^{\prime })-Q(s,a;\theta )\nabla Q(s,a;\theta )]$$
where L(θ) and TargetQ represent the loss function and objective function, respectively. ρ(·) denotes the probability distribution of choosing action a in a given environment s. At the iterative time step t + 1, the network weight parameters $\nabla_{\theta }L(\theta )$ are updated by two identical networks, namely the value network and the target network. To address the overestimation issue in Q-learning, a greedy strategy based on the deep double Q-network, which combines DQN with online network evaluation, is employed instead of using the target network for value estimation. The parameters are updated by Equation (4).
(4)$$Y_{t}^{DDQN}=r_{t+1}+\gamma Q(s_{t+1},\arg\underset{a}{\max}Q(s_{t+1},a;\theta_{t}))$$

### 3.2. Prioritized Experience Replay

The key to the prioritized experience replay mechanism lies in determining whether a sample is valuable or contributes to a larger TD-error (temporal difference error) [27]. The value of a sample increases as the error between the estimated value and the target value grows. Assuming the TD-error at sample i is defined as $\sigma_{i}$, the sampling probability can be defined as follows:(5)$$C_{i}=\frac{C_{i}^{j}}{\sum_{m}C_{m}^{j}}$$

The TD-error of each sample is represented by $C_{i}$ during calculation, and the significance of its error is modified by j. When j=1, the error value is used immediately; when j<1, the influence of samples with high TD-errors can be reduced, while the influence of samples with low errors is appropriately increased. There are two different ways to define $C_{i}$: priority proportion $C_{i}=\left|\theta_{i}\right|+\varepsilon$ and priority-based sorting method $C_{i}=1/rank(i)$, with rank(i) obtained through sorting $\left|\theta_{i}\right|$. When using the probability distribution of prioritized replay, the samples are drawn with unequal probabilities. Since the distributions of samples and action value functions are not identical, the model updates are biased. To correct this bias, the important sampling weights ω are used, as shown in Equation (6).
(6)$$\omega_{\alpha }={\left(\frac{1}{N\cdot P(\alpha )}\right)}^{\beta }$$

Here, *N* represents the number of samples stored in the experience replay buffer, and β denotes the correction factor. A weighted $\omega_{\alpha }$ is added before each learning sample to ensure unbiased updates. Different samples in the experience replay buffer have varying impacts on backpropagation due to different TD-errors. A larger TD-error results in a greater impact on backpropagation, while samples with smaller TD-errors have minimal influence on the calculation of the backward gradient. In the Q-network, the TD-error refers to the gap between the *Q*-values calculated by the target Q-network and the current Q-network, respectively. Therefore, based on the absolute value of the TD-error $\left|\delta_{t}\right|$ for each sample, the priority of that sample is proportional to $\left|\delta_{t}\right|$.

The SumTree binary tree structure is employed to store samples in the prioritized experience replay buffer [28]. The samples with larger absolute TD-errors are more likely to be sampled, leading to faster convergence of the algorithm. All experience replay samples are stored only in the lowest-level leaf nodes, with each node containing one sample, and the internal nodes do not store sample data. In addition to storing data, the leaf nodes also maintain the priority of each sample. The internal nodes, on the other hand, store the sum of the priority values of their child nodes, as illustrated by the numbers displayed on the internal nodes in Figure 1.

### 3.3. Reward Reshaping

Sparse reward signals are a series of rewards generated through the interaction between robot and environment, where most of the rewards obtained are non-positive, making it difficult for learning algorithms to associate a long series of actions with future rewards. Thus, the robot may never find a reward signal while performing a grasping operation, thus not learning how to execute a given task. It is assumed that a grasping operation will receive a higher reward value, such as 10, when the allowable error between the position of the end-effector and the target position reaches a certain value. During this process, only a small reward, such as -0.01, will be received at each step when the desired goal is not achieved. The determination of rewards is related to the adaptive size of the target, which can be expressed as:(7)$$r_{t}=\left\{\begin{array}{c} \begin{array}{cc} 10 & \left\|X_{\theta_{t}}-X^{T}\right\|\leq \rho (e) \end{array}\\ \begin{array}{cc} -0.01 & \;\left\|X_{\theta_{t}}-X^{T}\right\|>\rho (e) \end{array} \end{array}\right.$$

However, it is difficult to fully train the learning policy due to the scarcity of target rewards. When the end-effector and the target point are separated by a specific distance, the rewards are modified and intermediate rewards are adjusted. The reward setting is shown in Equation (8).
(8)$$r_{st}=\frac{\left\|X_{\theta_{t-1}}-X^{T}\right\|-\left\|X_{\theta_{t}}-X^{T}\right\|}{\left\|X_{\theta_{t}}-X^{T}\right\|}$$

In this context, $r_{st}$ must remain stable within the range of [−0.08, 0.08], as it represents the reward determined by the reward modification at step t in the above equation. If the magnitude of the intermediate reward is too large, it can affect the stability of the training process.

## 4. Push-Grasp Policy Design

This section designs a dexterous push-grasp combination strategy based on the visual attention mechanism in the case of sparse environmental rewards. The policy framework is visually illustrated in Figure 2.

### 4.1. Affordance Perception

Firstly, given an intermediate feature map $F\in R^{C\times H\times W}$ as input, the convolutional block attention module (CBAM) sequentially infers a one-dimensional channel attention map $M_{c}\in R^{C\times 1\times 1}$ and a two-dimensional spatial attention map $M_{s}\in R^{1\times H\times W}$. As shown in Figure 3, this module consists of two sequential sub-modules: the channel attention module and the spatial attention module [29]. The intermediate feature map is adaptively extracted through CBAM for each convolutional block of the deep network.

The entire attention mechanism process can be summarized as Equation (9).
(9)$$\begin{array}{l} F^{\prime }=M_{c}(F)⊗F\\ F^{\prime \prime }=M_{s}(F^{\prime })⊗F^{\prime } \end{array}$$
where ⊗ represents the element-wise multiplication. During the multiplication process, the attention values are propagated or replicated accordingly: the channel attention values are propagated along the spatial dimension, and vice versa. $F^{\prime \prime }$ is the output of the final feature extraction. The calculation process of the channel and spatial attention maps is shown in Figure 4. The channel attention submodule utilizes the outputs of both max pooling and average pooling from a shared network, while the spatial attention submodule utilizes two similar outputs pooled along the channel axis and passes them through a convolutional layer. The channel attention map is generated by leveraging the inter-channel relationships of the features. Since each channel of the feature map is treated as a feature detector, channel attention focuses on the given input image. To effectively compute channel attention, the spatial dimensions of the input feature map are compressed, and spatial information is aggregated using average pooling. Max pooling collects important information about different object features, enabling the inference of more fine-grained channel attention. Therefore, the simultaneous use of average pooling and max pooling features greatly enhances the representational capacity of the network.

In the channel attention module, spatial information of the feature map is first aggregated using average pooling and max pooling to generate two different spatial context descriptors: $F_{avg}^{c}$ and $F_{\max}^{c}$, representing the average-pooled features and max-pooled features, respectively. Then, the two descriptors are input into a shared network to produce the channel attention map $M_{c}\in R^{C\times 1\times 1}$. The shared network consists of a multi-layer perceptron (MLP) with one hidden layer. To reduce parameter, the hidden activation size is set to $R^{C/r\times 1\times 1}$, where r is the compression ratio. After applying the shared network to each descriptor, the output feature vectors are merged using element-wise summation. In summary, the computation of channel attention is shown in Equation (10).
(10)$$\begin{array}{l} M_{c}(F)=\sigma (MLP(AvgPool(F))+MLP(MaxPool(F)))\\ \;\;\;=\sigma (W_{1}(W_{0}(F_{avg}^{c}))+W_{1}(W_{0}(F_{\max}^{c}))) \end{array}$$
where σ represents the sigmoid activation function, $W_{0}\in R^{C/r\times C}$, and $W_{1}\in R^{C\times C/r}$. The MLP weight coefficients $W_{0}$ and $W_{1}$ are shared between the two inputs, and $W_{0}$ follows the ReLU activation function. The choice of the ReLU activation function is due to its nonlinear nature, which maps any input value to a non-negative output, thereby enhancing the expressive capacity of the neural network. Moreover, the sparsity and fast computation speed of the ReLU activation function make it particularly effective when processing large-scale image data. It can effectively prevent the problem of gradient vanishing.

The spatial attention module generates a spatial attention map based on the spatial relationships between features. Unlike the channel attention module, spatial attention focuses on identifying the effective information regions, complementing the channel attention. To compute spatial attention, average pooling and max pooling operations are first applied along the channel axis, and the feature descriptors are concatenated to create an effective feature representation. Applying pooling operations along the channel axis has been proven effective in highlighting informative regions [30]. On the concatenated feature descriptor, a convolutional layer is utilized to generate the spatial attention map $M_{s}(F)\in R^{H\times W}$, which encodes the locations to emphasize or suppress. By aggregating the channel information of the feature map using two pooling operations, two 2D maps, $F_{avg}^{s}\in R^{1\times H\times W}$ and $F_{\max}^{s}\in R^{1\times H\times W}$, are generated, representing the average-pooled and max-pooled features across channels, respectively. These are then concatenated and passed through a standard convolutional layer to produce the 2D spatial attention map. The computation of spatial attention is shown in Equation (11).
(11)$$\begin{array}{l} M_{s}(F)=\sigma (f^{7\times 7}([AvgPool(F);MaxPool(F)]))\\ \;\;\;=\sigma (f^{7\times 7}([F_{avg}^{s};F_{\max}^{s}])) \end{array}$$
where $f^{7\times 7}$ denotes the convolution operation with a 7 × 7 kernel.

The parameters of the visual attention network structure constructed in this section are shown in Table 1. The attention architecture (CBAMNet) in this paper is a convolutional block attention module, primarily based on the deep residual network (DenseNet-121). This network includes a convolutional layer and four attention blocks. The spatial attention and channel attention are employed in the residual cascade within the attention blocks. On one hand, a channel attention map is generated to direct attention towards global information; on the other hand, separate attention is paid to spatial feature maps of both the attention space and the target space. The two modules calculate complementary attention independently of each other and are combined sequentially to enhance attention to the position and feature information of objects in the workspace.

### 4.2. Maximum Entropy DQN

Assuming the action strategy is π, given N actions and corresponding reward vectors χ, the entropy regularization strategy optimization problem is defined as follows:(12)$$\max\left\{\pi \chi +\eta E(\pi )\right\}$$

The degree of exploration is controlled by η, and η≥0.

The most important issue in reinforcement learning is exploration-exploitation. Entropy of policy is defined as follows:(13)$$E(\pi )=-\sum_{\pi^{\prime }\in \pi }\pi^{\prime }\log(\pi^{\prime })$$

The entropy of deterministic policy is relatively low, and the entropy of random policy is relatively high. The optimal solution for the maximum entropy objective is obtained through the Soft Bellman equation, as shown in Formula (14).
(14)$$Q(s_{t},a_{t})=E[r_{t}+\gamma soft\max_{a}Q(s_{t+1},a)]$$
(15)$$soft\max_{a}f(a)=\log\int expf(a)da$$

Combining the Formula (12), the larger η, the more entropy becomes dominant and tends towards a random strategy (exploration); when η is smaller, the reward is dominant and tends towards deterministic strategies (exploitation). By mapping a reward vector into an uncertain strategy, the component of the vector χ is the probability of selecting that action.

The input of the DQN network is the state vector φ(s) corresponding to the state s, and the output is the action-value function Q for all actions under that state. Two neural networks with identical structures are constructed: the MainNet, which continuously updates the current neural network parameters, and the TargetNet, which is used to update the Q-value. The objective function is defined as:(16)$$TargetQ=r+\gamma max_{a\prime }Q(s^{\prime },a^{\prime };\theta )$$

The loss function of the DQN network is defined as:(17)$$L(\theta )=E[{\left(TargetQ-Q(s,a;\theta )\right)}^{2}]$$
where θ represents the neural network parameter. Gradient descent is employed to approximate the current *Q*-values to the target *Q*-values. The gradient update as shown in Formula (18).
(18)$$\theta_{t+1}=\theta_{t}+\alpha [r+\gamma \underset{a^{\prime }}{max}Q(s^{\prime },a^{\prime };\theta )-Q(s,a;\theta )]\nabla Q(s,a;\theta )$$

To reduce data correlation, the neural network approximates the value function by calculating the TD target network represented as $\theta^{-}$, and the network used for approximating the value function is represented as θ. The network for approximating the action-value function is updated at each step, and the update process is as follows:(19)$$\theta_{t+1}=\theta_{t}+\alpha [r+\gamma \underset{a^{\prime }}{max}Q(s^{\prime },a^{\prime };\theta^{-})-Q(s,a;\theta )]\nabla Q(s,a;\theta )$$

By combining a random policy with the DQN network, and integrating a visual attention feature extraction network model with an action network model, the action Q-values is predicted. The priority sampling is conducted based on the prioritized experience replay mechanism. The loss function is defined as:(20)$$\frac{1}{m}\sum_{j=1}^{m}\omega_{j}{\left(y_{i}-Q(\varphi (S_{j}),A_{j},\omega )\right)}^{2}$$
where $\omega_{j}$ represents the priority weight of the *j*-th sample, which is normalized from the TD error $\left|\delta_{t}\right|$. After gradient updating the parameters of the Q-network, the TD error needs to be recalculated and updated on the SumTree. The gap between them is the entropy of the policy. When η→0, the entropy regularized policy optimization problem becomes the standard expected reward objective, where the optimal solution is the hard-max policy.

The output features are fused and fed into the ME-DQN network (as shown in Figure 5) to generate affordance maps for grasping actions. A greedy strategy is employed to obtain pixel-wise predicted Q-values and action probabilities. The self-supervised training is aim to achieve a superior target value, as described in Formula (21).
(21)$$Q_{i+1}(s_{t},a_{t})=R_{t+1}(s_{t},s_{t+1})+\gamma \max_{a}Q(s_{t+1},a;\theta_{t+1})$$
where $Q_{t+1}$ represents the predicted value of executing an action, $R_{t+1}(s_{t},a_{t})$ is the reward value obtained after executing action $a_{t}$, and $\theta_{t+1}$ denotes the network parameters at time *t* + 1. The maximum predicted *Q*-value is achieved by selecting the optimal action, and the Q-function in the network indicates the degree of advantage or disadvantage for the robot to execute an action in state *s*. The prioritized experience replay improves the decision-making process, with the advantage function representing the behavioral performance of the robot. The ME-DQN divides the Q-network into two parts: the first part is only related to the state s and is independent of the specific action *a*, defined as the value function V(s,w,α); the second part is related to both the state and the action, with the advantage function defined as A(s,a,w,β). The state-action value function is derived from this, as shown in Equation (22).
(22)Q(s,a,w,α,β)=V(s,w,α)+A(s,a,w,β)
where *w* represents the network parameters, α denotes the network parameters for the value function, and β represents the advantage function parameter. The advantage function determines whether the current action yields a higher reward value compared to other actions, and the priority sorting gets rid of unimportant experience sequences. Meanwhile, the trouble of adjusting hyper-parameters is eliminated.

## 5. Experiment Analysis

In this section, a comparative analysis is conducted on the grasping performance of single target objects and multi-objective unknown objects. The effectiveness and generalization ability of the algorithm are verified through simulation and real experiments.

### 5.1. Experimental Setup

To reduce robot wear and tear, similar to the simulation environment of Zeng et al. [11], a simulation experiment platform was built based on V-REP [31], with its internal inverse kinematics module used for robot motion planning and Bullet Physics for dynamics. The simulation environment incorporates a UR5 robotic arm and a two-finger parallel gripper (RobotIQ 2F-85), with the adjustable range of the gripper being 0–85 mm. The deep camera selected is the RealSense D435i, with a resolution of 1280 × 720. The graphics card model is NVIDIA RTX 2080 Ti, and the operating hardware consists of a 3.2 GHz CPU and 64G of memory. The operating system is Ubuntu16.04, and the programming language is Python. The libraries used include OpenCV, Numpy, Pandas, and others. The physical experiments in this section are based on the JAKA Zu 7 six-axis robotic arm, with the two-finger gripper being the WHEELTEC.

### 5.2. Training

The heightmap is constructed by capturing visual 3D data from an RGB-D camera statically mounted at the end of the robotic arm and orthogonally projecting it onto the RGB-D heightmap. The heightmap is rotated in 16 directions to enhance data utilization. A spatial-channel attention is to improve the expression of objects and extract workspace features. After the completion of the action network, an affordance map of the object is generated to further enhance its expressiveness. Combined with the dense pixel maps predicted by a fully convolutional network based on DenseNet-121, several optional locations are identified. The decision system determines the optimal grasp point based on the magnitude of the *Q*-value, with Q<0.5 indicating unsuitability for grasping in the experiments. To avoid local optimal solutions, an *ε*-greedy strategy is employed to randomly execute grasping actions for exploration.

There are several objects randomly being placed on a workspace scenario measuring 0.8 m × 0.65 m in training. The iterative training is conducted for 10,000 epochs, with a maximum of 10 operations performed in each scenario. The exploration rate discount factor is set to 0.99, and the momentum coefficient is set to 0.95. Network parameters are updated based on stochastic gradient descent. Due to insufficient sample data, training begins once the number of sequential samples stored in the replay buffer reaches 5000. The maximum memory capacity is set to 580,000. The ReLU activation function, batch normalization, and dropout (ranging from 0.2 to 0.4) are added after each layer. The optimizer is Adam, with a learning rate of 10^−4^.

At the initial stage of training, the ε-greedy strategy is employed for continuous exploration and exploitation, aiming to find the optimal policy to guide the robot to execute the best actions. As shown in the grasping reward curve depicted in Figure 6, the initial stage exhibits low values for both the current state grasping reward and the average reward due to the limited number of data tuples in the experience replay buffer. As the training proceeds, the prioritized experience replay is utilized to reduce data correlation. This involves pixel-by-pixel prediction of the value function V(s,w,α) and the action execution probabilities. The mean squared error loss function based on sample priority is then used to update all parameters *w* of the Q-network through gradient backpropagation in the neural network. Combined with the advantage function, the optimal state-action value function Q(s,a,w,α,β) is obtained. The reward function gradually converges after 8000 iterations. It indicates that the model has stabilized. This ensures that the robot can reliably execute optimal grasping actions based on the learned representations and policies.

### 5.3. Object Grasping Simulation Experiments

The experiment is conducted in a same experimental environment for the ME-DQN network using three different backbones (DenseNet-121, DenseNet-169, and DenseNet-201). In the vrep simulation environment, a single object was dropped in each iteration, and a total of 50 unknown objects with various structural types, including cubes (cub), cylinders (cy) and others, were set up for grasping operations (see Figure 7). The number of grasping attempts in each scenario was limited to no more than three. Among testing, the architecture based on DenseNet-121 exhibited the most prominent performance in terms of grasping success rate (GS), grasping efficiency (GE), and the time required to grasp each object (GT). Specifically, the DenseNet-121-based model achieved a 100% grasping success rate.

Evaluation was conducted by comparing three metrics as summarized in Table 2. The results indicate that the DenseNet-121 backbone is particularly suitable for the task of object grasping in the given simulation environment, offering high accuracy and efficiency. This may be attributed to the ability of DenseNet-121 to extract rich and discriminative features from input data, enabling the network to effectively identify and locate objects for successful grasping.

The dense object grasping experimental scenarios are categorized into two types: identical structure and different structure, as shown in Figure 8 and Figure 9. In simulation environment, 10 objects are randomly generated in each round of the experiment, and the number of grasping attempts per task is limited to less than 30. A reward value of 10 is obtained when the end-effector is successful grasping. Only a small reward of −2 is received for each step if not. To avoid local optimal solutions, an ε-greedy exploration strategy is adopted, which attempts to take random actions with a certain probability to explore better policy instead of blindly selecting the action with the best value based on the current policy. We initialize ε as 0.99 and gradually reduce it to 0.01 during the training process.

The training results of multi-object grasping based on different backbones with various structures are presented in Figure 10. The grasping success rate curve of the active Deep Q-Network model based on the DenseNet-201 architecture rapidly rises in the initial stage but reaches saturation early on. The other two algorithms show a slower increase at the beginning but present stable performance later on. After 2000 iterations, the grasping success rates of the three algorithms are 52% (red), 38% (green), and 40% (blue), respectively. All three algorithms converge with average grasping success rates reaching 51% (red), 67% (green), and 92% (blue) conducting 4000 iterations. Although the DenseNet-201-based achieves the fastest speed and the DenseNet-169-based demonstrates a better balance in the later stage, the method (DenseNet-121-based) proposed in this paper exhibits a higher grasping success rate in the long run. This is mainly due to the fact that the DenseNet-121 network has fewer parameters and depth, which alleviates the issue of gradient vanishing while enhancing the information transmission of feature maps.

A deep analysis of the grasping performance in two types of scenarios is presented in Table 3. The proposed method in this paper exhibits a significant decrease in grasping efficiency for objects with different structures, while the change in success rate is relatively insignificant. This is primarily due to the fact that objects with different structures lack specific contour features and contain less semantic feature information. Consequently, the action network requires greedy exploration and exploitation during the object grasping process. The action network must extensively explore various grasping policy to identify the optimal grasping approach for each unique object structure, leading to a decrease in overall grasping efficiency. However, the success rate remains relatively stable as the model is able to adapt and learn effective grasping skills for a wide range of object shapes and sizes.

For all benchmarks, we conducted 4000 iterations of training to demonstrate that the overall performance of our proposed method outperforms others. The simulations incorporate the utilization of 10 different 3D toy blocks, wherein their shapes and colors are randomly selected during the experiments. As illustrated in Figure 11, after approximately 2500 iterations of training, the grasping success rate of ME-DQN stands at around 80%. Following further training, the performance after 4000 iterations reaches approximately 93%. In the early stages, the training performance of Dual viewpoint [14] and VPG [11] is higher than ME-DQN, mainly due to the fact that ME-DQN incorporates pushing actions into its training from the beginning, increasing the exploration of complex environments, thus resulting in lower performance initially. In contrast, the Rule-based [15] and VPG-only depth [30] employs a greedy strategy in the early stages, selecting the maximum predicted Q-value. During this phase, the grasping prediction value is slightly higher than the pushing prediction value, and the impact of environmental noise is minimal, leading to a higher grasping success rate. However, as the environmental noise increases significantly in the later stages, after 3000 iterations, the success rate of Rule-based and VPG-only depth falls below 75%, while ME-DQN maintains a success rate of around 93%.

We conducted 20 separate trials for unknown objects, with each trial capped at a maximum of 30 action attempts. As shown in Table 4 and Figure 12, the test results indicated significant variations in success rates and action efficiency among the different algorithms. We found that VPG-only depth [30] and VPG [11] tends to push objects towards the edges or even corners, a behavior that diminishes grasping success rate. In contrast, a dual viewpoint [14] ensures that the entire grasping process is more suited to the random environment with unknown objects. However, the arrangement structure of unknown objects differs from that of the objects found in the training set, which occasionally results in exploring consumption or failed pushing attempts. The rule-based method [15] heavily relies on find the best grasp rectangle based on image and is more possible to treat multiple objects as single object. Therefore, the grasping success rate performs the worst among all baseline methods. Specifically, our method demonstrated a consistently high success rate and completion across a wide range of object shapes, while others performed poorly in common scenarios.

### 5.4. Ablation Experiment

As shown in Table 5, a statistical analysis was conducted on the training iterations required for the multi unknown object grasping success rates to reach 60%, 70%, 80%, and 90% in the ablation experiment. Without the advantage function and attention-based object affordance perception network, the grasping success rate of the DQN (DenseNet121) was below 80%. Lacking maximum entropy regularization, it relied more on existing data and policy, and seldom attempted unknown actions during the interaction with different grasping actions and the environment. The ME-DQN-noAF model without the advantage function increased the variance during the learning process. In the case of multi-object with limited resources, it required more time to distinguish the effects of different actions. If the state space and action space were large, the number of active exploration steps would increase significantly, making it difficult for the algorithm to learn the optimal policy in a short time. Ignoring the attention mechanism, the ME-DQN-noattention model was unable to focus on the important parts of the input information, resulting in reduced efficiency and decision-making accuracy during the learning process, as well as decreased generalization ability. Finally, the ME-DQN model proposed in this paper reduced the interference of irrelevant information, enabling the model to focus more on the most important factors for the current task. As a result, a high grasping success rate of 91.6% could be achieved after 711 attempts.

### 5.5. Physical Experiment

The simulation experiments provide a controllable and safe environment for testing and adjusting grasping algorithms, while real-world scenarios possess higher complexity and unpredictability. Transferring simulation experiments to real-world settings can assist robots to learn how to cope with these challenges, such as lighting conditions, physical disturbances, and complex backgrounds, as shown in Figure 13.

In each grasping attempt, the network receives visual signals from the depth camera. Figure 13a,d,g,j are original states. Figure 13b,e,h,k represent pushing actions. Figure 13c,f,i,l are successful grasping, with each scene executing no more than twice as many actions as the object to be grasped. To validate the effectiveness of the proposed algorithm in real world scenarios, three types of unknown object grasping experiments were conducted with 10, 20, and 30 objects, respectively. As shown in Table 6, the algorithm proposed in this paper achieved an average grasping success rate of approximately 91.6% with 511 grasping attempts, significantly outperforming the other three methods. This demonstrates its potential for generalization to grasping operations of unknown objects in cluttered environments. Even when grasping operations were performed on a larger number of new objects (30 objects), a grasping success rate of 87.2% could still be achieved. The attention mechanisms and prioritized experience replay reduced the number of random predicted grasps, significantly improving grasping efficiency. It is difficult to obtain external environmental parameters such as friction coefficient, centroid, and spring coefficient in the simulation environment. Besides, the motor control in the real-world experiments has certain precision errors. The main reason for the difference in success rate is that the dynamic model of robot in the real environment is difficult to be as accurate and stable as that in simulation. In addition, objects are randomly placed in the simulation environment, while objects are closely arranged in real-world, leading to the increase of interference factors and the difficulty of reasoning decision-making. Overall, the grasping success rate in real world experiments is generally lower than that in simulation experiments.

## 6. Conclusions

This paper proposes a maximum entropy Deep Q-Network for dexterous grasping of multiple unknown objects based on the attention mechanism. In unstructured scenes, the robot grasping operations are modeled using Markov decision processes. The object affordance perception based on spatial-channel attention allows the robot to dynamically adjust the focus to adapt to environmental changes and learn more generalized feature representations, especially with strong generalization ability when facing diverse and unknow objects. A prioritized experience replay mechanism is designed to deal with the high-dimensional perceptual inputs and complex decision tasks, reducing reliance on a large amount of similar and low-value repetitive redundant data. Two neural networks with the same structure are constructed. In the environments with sparse rewards, reward reshaping during the exploration phase guides the robot to conduct more efficient exploration, especially accelerating the learning process when approaching the object. The effectiveness of the method is validated through quantitative experiments and comparative analysis on single-object and multi-object grasping in unstructured environments. The simulation environment is also transferred to real world for experiments to more accurately evaluate the performance of robot grasping. As a future research direction, this study can be extended to explore grasping in scenes with multiple unknown objects such as adhesion and stacking.

## Figures and Tables

**Figure 1 entropy-26-00416-f001:**
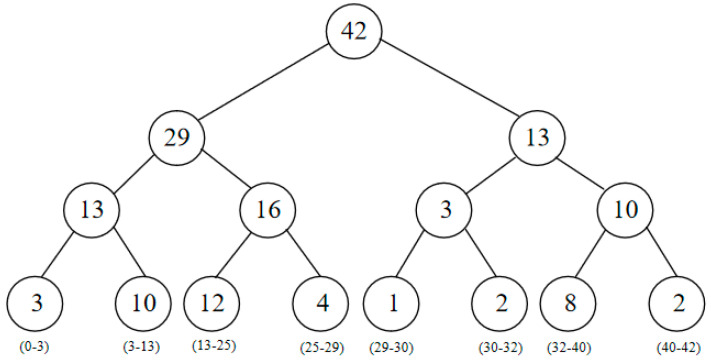
Priority sampling and storage based on SumTree structure.

**Figure 2 entropy-26-00416-f002:**
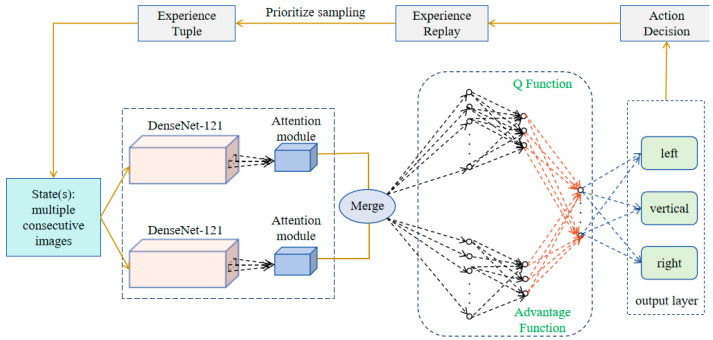
The policy framework of robot dexterous grasping.

**Figure 3 entropy-26-00416-f003:**
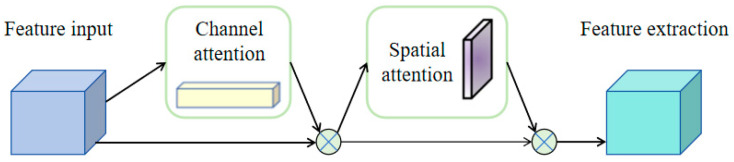
Convolutional attention mechanism block.

**Figure 4 entropy-26-00416-f004:**
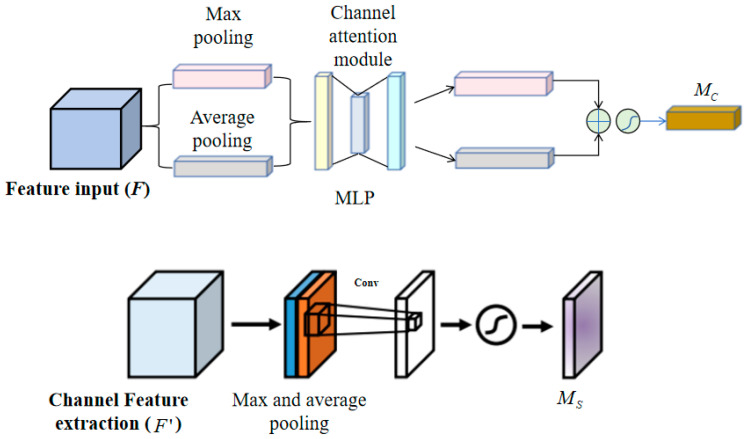
The schematic diagram of channel attention module and spatial attention module.

**Figure 5 entropy-26-00416-f005:**
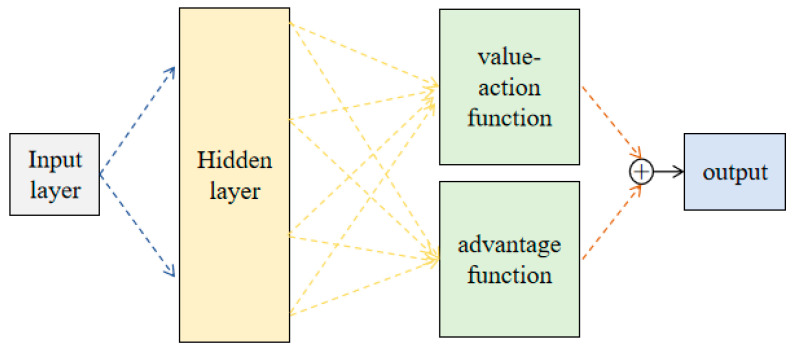
The schematic diagram of maximum entropy DQN network.

**Figure 6 entropy-26-00416-f006:**
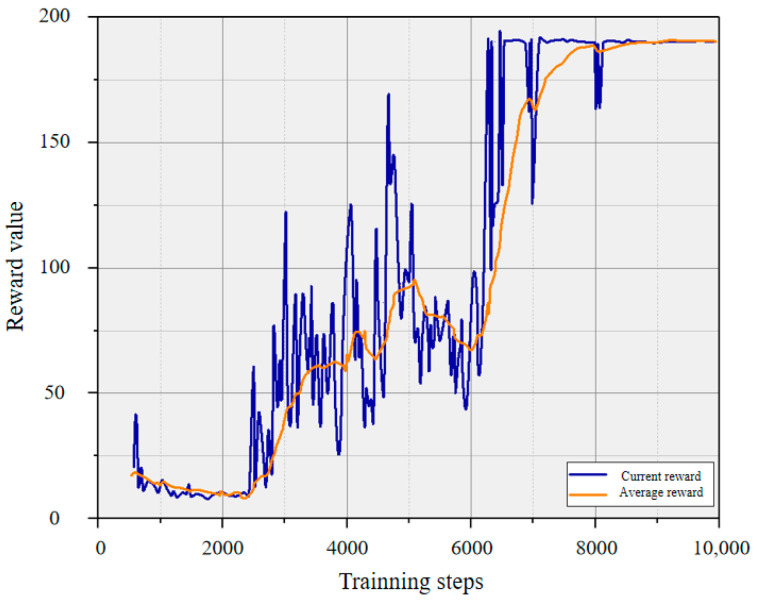
The reward value curve for grasping actions.

**Figure 7 entropy-26-00416-f007:**
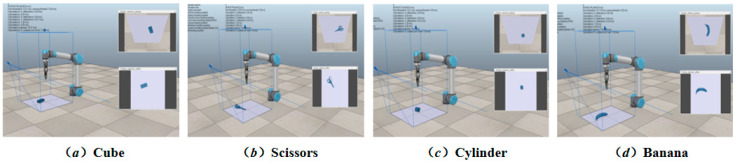
The schematic diagram of ME-DQN network.

**Figure 8 entropy-26-00416-f008:**
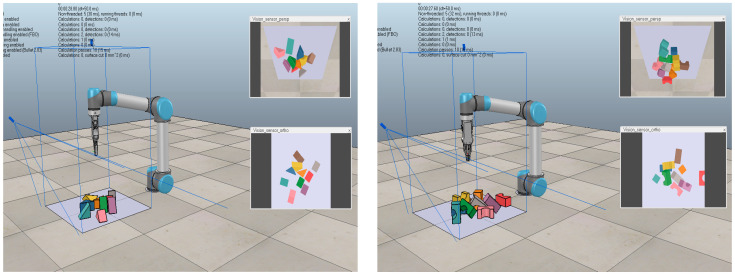
The same structure.

**Figure 9 entropy-26-00416-f009:**
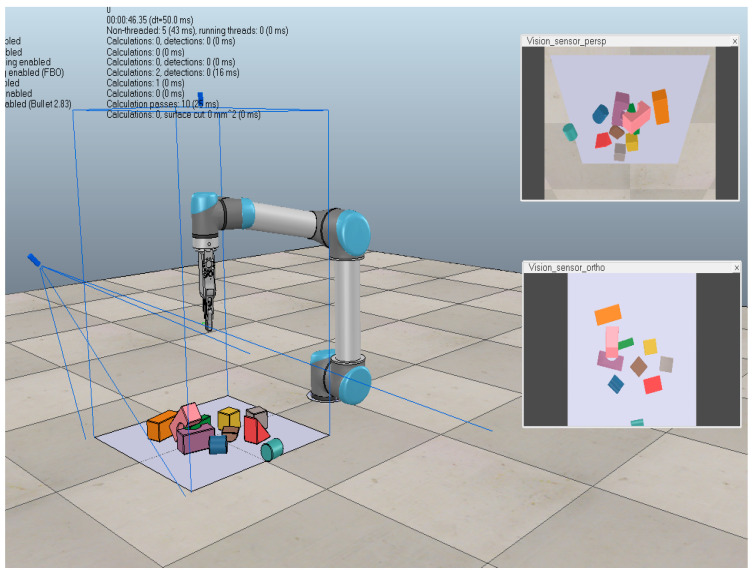
The different structure.

**Figure 10 entropy-26-00416-f010:**
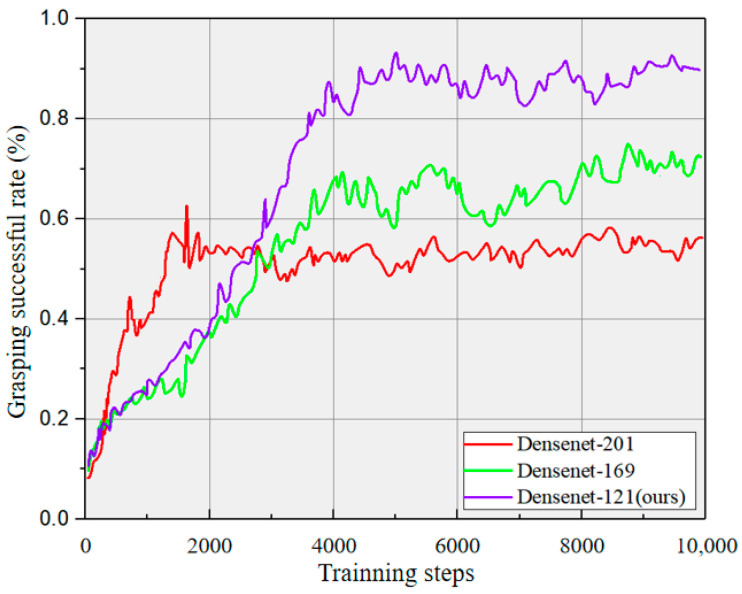
The training results for a multi-object with different structures based on different backbones.

**Figure 11 entropy-26-00416-f011:**
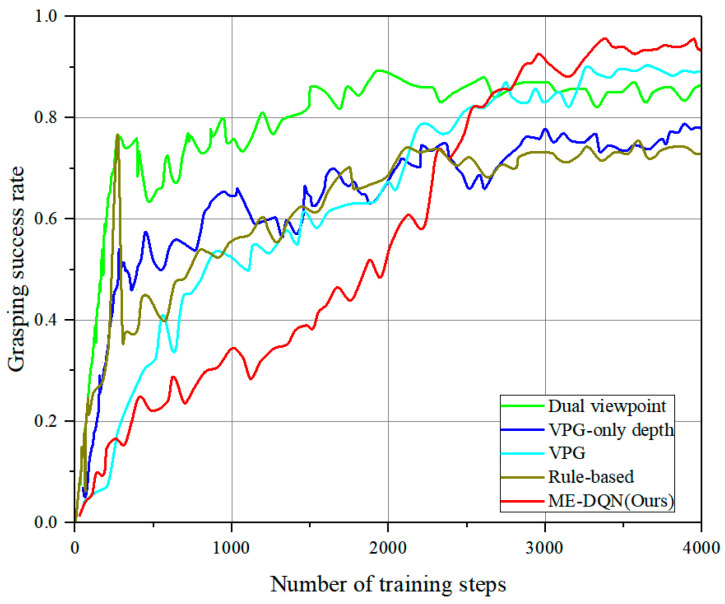
The comparison of training for novel unknown objects with benchmarks in simulation.

**Figure 12 entropy-26-00416-f012:**
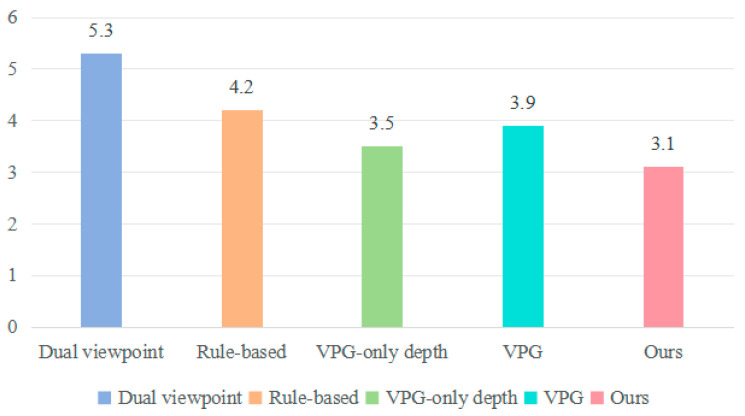
The evaluation of mean action efficiency.

**Figure 13 entropy-26-00416-f013:**
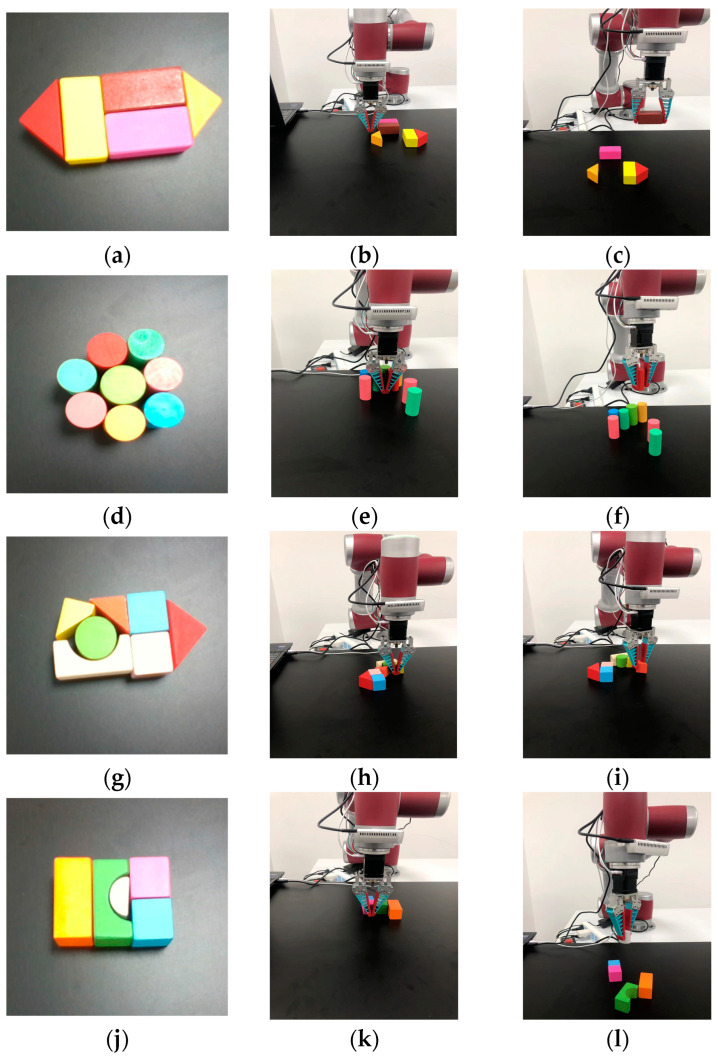
The grasping experiments of multiple unknown objects in real world.

**Table 1 entropy-26-00416-t001:** The parameters of the visual attention network.

Layer Name	Output Size	Kernel Size/Number	Output Feature Maps
Conv	112 × 112		64
Pooling	56 × 56		64
Attention block_1	56 × 56	$\left[\begin{array}{cc} 1\times 1 & \mathrm{conv}\\ 3\times 3 & \mathrm{conv} \end{array}\right]\times 6$	256
Transition layer_1	56 × 56	$\begin{array}{cc} 1\times 1\times 128 & \mathrm{conv} \end{array}$	128
	28 × 28		125
Attention block_2	28 × 28	$\left[\begin{array}{cc} 1\times 1 & \mathrm{conv}\\ 3\times 3 & \mathrm{conv} \end{array}\right]\times 12$	512
Transition layer_2	28 × 28	$\begin{array}{cc} 1\times 1\times 256 & \mathrm{conv} \end{array}$	256
	14 × 14		256
Attention block_3	14 × 14	$\left[\begin{array}{cc} 1\times 1 & \mathrm{conv}\\ 3\times 3 & \mathrm{conv} \end{array}\right]\times 24$	1024
Transition layer_3	14 × 14	$\begin{array}{cc} 1\times 1\times 512 & \mathrm{conv} \end{array}$	512
	7 × 7		512
Attention block_4	7 × 7	$\left[\begin{array}{cc} 1\times 1 & \mathrm{conv}\\ 3\times 3 & \mathrm{conv} \end{array}\right]\times 16$	1024

**Table 2 entropy-26-00416-t002:** The grasping evaluation of single object based on different backbone.

	GS(%)			GE (Number per Hour)			GT (s)		
Module	cub	cy	o	cub	cy	o	cub	cy	o
DenseNet-201	78.5	75.1	68.5	800	642	590	4.5	5.6	6.1
DenseNet-169	89.2	85.7	80.3	947	734	679	3.8	4.9	5.3
DenseNet-121(Ours)	100	100	100	972	782	750	3.7	4.6	4.8

**Table 3 entropy-26-00416-t003:** The comparison of grasping performance between two types of scenes.

	GS (%)	GE (Number per Hour)	GT (s)
Same structure	93.1	702 ± 3	7.9
Different structure	92.4	519 ± 3	10.8

**Table 4 entropy-26-00416-t004:** Test results for unknown objects.

	Evaluation Metrics (Mean %)	
Methods	Completion	GS (%)
Dual viewpoint [14]	92	83.2
Rule-based method [15]	90	72.8
VPG-only depth [30]	96	74.6
VPG [11]	90	86.9
Ours	98	92.4

**Table 5 entropy-26-00416-t005:** The ablation experiments on multiple unknown objects.

Module	60%	70%	80%	90%
DQN (DenseNet121)	185	525	-	-
ME-DQN-noAF	269	337	402	-
ME-DQN-noattention	213	286	592	-
ME-DQN (ours)	287	368	435	711

**Table 6 entropy-26-00416-t006:** The comparative experiments on real unstructured complex stacking scenes.

Methods	Attempts	Average Successful Rate/Individual Object Time	Successful Rate of Empty Workplace		
			10 Objects	20 Objects	30 Objects
UCB [32]	523	82% (15.8 s)	89%	83%	75%
3DCNN [33]	471	87% (12.7 s)	92.5%	89.5%	79%
Coordinator [34]	509	85% (17.3 s)	94.5%	81%	79.5%
VPG [11]	497	82.9% (10.9 s)	94.8%	83.6%	70.3%
Ours	511	91.6% (8.9 s)	96%	88%	87.2%

## Data Availability

The data presented in this study are available on request from the corresponding author.

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
