# Peer review of "Towards Multi-Objective Object Push-Grasp Policy Based on Maximum Entropy Deep Reinforcement Learning under Sparse Rewards"

_entropy, 2024, doi:10.3390/e26050416_

Round 1
Reviewer 1 Report
Comments and Suggestions for Authors
In this paper, a maximum entropy Deep Q-Network for dexterous grasping of multiple unknown objects based on the attention mechanism is presented. Also, this paper merged Fully Convolutional Networks' robust feature extraction capabilities with the efficient feature selection of the attention mechanism across diverse task scenarios. For this purpose, training and experimental processes were carried out successfully.
The authors stated that the reason why the comprehension success rate in real-world experiments is lower than in simulation experiments is because the objects are irregular and diverse. However, upon closer examination of both simulation and real-world experiments, it becomes evident that the objects share similarities. The author should provide explanations for other potential factors contributing to the difference in success rates between simulation and real-world experiments.
Author Response
Dear Editor and Reviewers:
First of all, we would like to thank you for reviewing the manuscript. We thank you for the time taken to read the document. Your comments are very instructive and inspired us a lot to work better. We have revised our manuscript after carefully reading your comments, and all the changes have been highlighted with different colors. Our response to each comment can be found in the following list of items.
We hope that the revision will improve the paper and the changes made in attempts to address your points are satisfactory. We would be glad to respond to any further questions and comments that you may have.
Response to the Comments of Reviewer # 1
Comment 1: In this paper, a maximum entropy Deep Q-Network for dexterous grasping of multiple unknown objects based on the attention mechanism is presented. Also, this paper merged Fully Convolutional Networks' robust feature extraction capabilities with the efficient feature selection of the attention mechanism across diverse task scenarios. For this purpose, training and experimental processes were carried out successfully.
Response: Thanks for the reviewer’s summary and positive comments on our paper.
Comment 2: The authors stated that the reason why the comprehension success rate in real-world experiments is lower than in simulation experiments is because the objects are irregular and diverse. However, upon closer examination of both simulation and real-world experiments, it becomes evident that the objects share similarities. The author should provide explanations for other potential factors contributing to the difference in success rates between simulation and real-world experiments.
Response: In fact, objects in the simulation environment are very similar to those in the real experiment. It is difficult to obtain external environmental parameters such as friction coefficient, centroid, and spring coefficient in the simulation environment. Besides, the motor control in the real-world experiments has certain precision errors. The main reason for the difference in success rate is that the dynamic model of robot in the real environment is difficult to be as accurate and stable as that in simulation. In addition, objects are randomly placed in the simulation environment, while objects are closely arranged in real-world, leading to the increase of interference factors and the difficulty of reasoning decision-making. We have made careful modifications. Please see Page 16, blue color, Section 5.5.

Reviewer 2 Report
Comments and Suggestions for Authors
Author Response
Dear Editor and Reviewers:
First of all, we would like to thank you for reviewing the manuscript. We thank you for the time taken to read the document. Your comments are very instructive and inspired us a lot to work better. We have revised our manuscript after carefully reading your comments, and all the changes have been highlighted with different colors. Our response to each comment can be found in the following list of items.
We hope that the revision will improve the paper and the changes made in attempts to address your points are satisfactory. We would be glad to respond to any further questions and comments that you may have.
Response to the Comments of Reviewer # 2
Comment 1: The current title does not effectively reflect the core contributions and the unique approach of the paper. While the concepts of "entropy" and "sparse rewards" are well-recognized in the reinforcement learning (RL) community, especially among researchers focused on cluttered or well-structured object environments in robotics, the title should highlight the novelty of the approach to better distinguish it from existing work.
Response: Thank you very much for your insightful comment. Currently, the robots need to confront a wide range of objects with diverse shapes, and often the specific instances of these objects are unknown in unstructured environments. Traditional methods (e.g. deep learning) rely on training with large-scale labeled sample data, but the data becomes sparse in environments with continuous and high-dimensional state spaces. This results in the trained models with weak generalization capabilities when transferred to real robot. In fact, this paper delves into robot perception and decision-making methodologies, aiming to achieve efficient and precise grasping of multi-objective objects (known or unknown) in unstructured environments. We have revised the abstract with red color in page 1.
Comment 2: The introduction requires substantial revisions to better focus on the specific advancements the authors wish to make within the field. The current flow of the introduction lacks coherence. The discussion of traditional robotic grasping techniques, deemed not challenging, should be omitted or minimized, given the predominance of AI-based methods in recent research. It would be beneficial for the introduction to pivot more directly towards AI-based grasping techniques. The introduction lacks citations to key studies that are foundational to this research area. For benchmarking purposes, significant papers such as those found in the review article "Sensors 2022, 22(20), 7938; [https://doi.org/10.3390/s22207938]" should be discussed to set the context for the current study's contributions and positioning relative to the existing literature.
Response: Thank you very much for your insightful comment. We have revised the introduction. Please see Page 1-2, Red color, Section 1.
Comment 3: Related Work Section: The Related Work section is crucial as it contextualizes the current study within the broader research landscape by discussing how other works have addressed similar issues through different methodologies. This paper, however, lacks sufficient references to key studies that form the foundation of the discussed methodologies. The authors should revisit the recommendations made in the introduction comments regarding essential references that need to be included to enhance the depth and relevance of the discussion.
Response: Thank you very much for your insightful comment. We have added the related work section to enhance the depth and relevance of the discussion.. Please see Page 2-3, Red color, Section 2.
Comment4: Lines 161-175: The categorization of deep reinforcement learning algorithms into three types is stated without citing any sources. It is imperative to reference original works where these classifications are discussed.
Response: Thank you very much for your insightful comment. We have added new references in the paper. Please see Page 4, Red color, Section 3.1.
Comment 5: Subsection 2.2. Prioritized Experience Replay: This subsection directly borrows from the work of Schaul et al., yet fails to acknowledge this source: [4]: Schaul, T., Quan, J., Antonoglou, I., & Silver, D. (2016). Prioritized Experience Replay. ArXiv:1511.05952 [Cs]. Available online.
Response: Thank you very much for your insightful comment. We have added new references in the paper. Please see Page 4, Red color, Section 3.2.
Comment 6: Figure 1: This figure is adapted from a publication by Jin et al., which has not been cited: [1]: Jin, Y.; Liu, Q.; Shen, L.; Zhu, L. Deep Deterministic Policy Gradient Algorithm Based on Convolutional Block Attention for Autonomous Driving. Symmetry 2021, 13, 1061. https://doi.org/10.3390/sym13061061.
Response: Thank you very much for your insightful comment. We have added new references in the paper. Please see Page 5, Red color, Section 3.2.
Comment 7: Section 3.1. Affordance Perception: The entire subsection appears to be derived from Woo et al.’s work, including figures, equations, and explanations, yet no citations are provided: [5]: Woo, S., Park, J., Lee, J. Y., & Kweon, I. S. (2018). CBAM: Convolutional Block Attention Module. In Proceedings of the European Conference on Computer Vision (ECCV) (pp. 3-19). Available online.
Response: Thank you very much for your insightful comment. We have added new references in the paper. Please see Page 6, Red color, Section 4.1.
Comment 8: Section 3.2. Maximum Entropy DQN: The equations used in this subsection lack citations, which is necessary to credit the original sources.
Response: Thank you very much for your insightful comment. The method mentioned in this section is a new algorithm designed by integrating reinforcement learning theory, and thus no citations are included.
Comment 9: Comparison with the Benchmark: The analysis of benchmarks within this paper is insufficient, particularly concerning the critical task of comparing performance against key studies that focus on grasping objects in cluttered environments. To strengthen the manuscript, it is essential to include comparative analysis with additional relevant works that have previously addressed similar challenges. Specifically, references [1], [10], and [12] from the provided references in the introduction comments are crucial for a comprehensive benchmark comparison. These references not only offer a diverse range of methodologies and results but also set a standard for evaluating the effectiveness and novelty of the proposed approach. Including these comparisons will significantly enhance the paper's relevance and credibility within the field, demonstrating how the proposed method stands in relation to established techniques.
Response: Thank you very much for your insightful comment. To verify the effectiveness of the algorithm, we select the recent researches for a comprehensive benchmark comparison. Please see Page 14, Red color, Section 5.3.
Comment 10: Simulation Environment: The simulation environment setup is taken from a project detailed on GitHub, yet no acknowledgment is made: o Original GitHub project: https://github.com/andyzeng/visual-pushing-grasping.
Response: Thank you very much for your insightful comment. We have provided new explanations in the simulation experiment. Please see Page 10, Red color, Section 5.1.
Comment 11: Additional References: The use of CoppeliaSim (formerly V-REP) in the simulations is not credited: [6]: Rohmer, M. F. E., Singh, S. P. N. (2013). V-REP: A Versatile and Scalable Robot Simulation Framework. IROS.
Response: Thank you very much for your insightful comment. We have added new references in the paper. Please see Page 10, Red color, Section 5.1.

Round 2
Reviewer 2 Report
Comments and Suggestions for Authors
I have a few comments for the authors:
-
Reference Accuracy (Lines 73-75): The paper by Zeng et al. [11], referenced in your manuscript, does not appear to accurately reflect the content it is cited to support. The cited study focuses on "grasp-and-place" using classical image matching for tactile grasping and reliable object grasping from clutter without precise placement requirements. However, this description better aligns with the following reference:
A. Zeng et al., "Robotic Pick-and-Place of Novel Objects in Clutter with Multi-Affordance Grasping and Cross-Domain Image Matching," 2018 IEEE International Conference on Robotics and Automation (ICRA), Brisbane, QLD, Australia, 2018, pp. 3750-3757, doi: 10.1109/ICRA.2018.8461044.
Please review the reference and ensure that it corresponds accurately to the claims and context in which it is used.
-
Clarification on Comparative Analysis (Line 110): The manuscript states, "More closely related to our work is that of Zeng et al. [11]." This implies a benchmark comparison with the method described by Zeng et al. However, it is unclear from the text whether this comparison is theoretical or involves actual training and testing under similar conditions. Additionally, it is not evident why the approach by Zeng et al. was not included as a benchmark in Table 4 alongside the other benchmarks ([29], [30], and [31]). For clarity and completeness, please specify whether Zeng et al.'s method was directly compared in your experiments, discuss the scenarios under which both methods were evaluated, and explain the rationale for including or omitting this work in the benchmark comparison presented in Table 4.
-
Inclusion of Benchmark Studies in Related Work: The benchmarks presented in the results (Table 4 for benchmarks [29], [30], and [31]) are not mentioned in the Related Work section. It is crucial for consistency and reader understanding that these benchmarks be discussed in the Related Work section if they are pivotal to the study’s evaluation.
-
Presentation of Training and Testing Results: The results for benchmarks [29], [30], and [31] only disclose testing outcomes. For a comprehensive evaluation, it is advisable to include both training and testing results. Presenting the training environment and methodology would provide a more reliable basis for comparing the efficacy of the different approaches tested.
Please consider these points to enhance the clarity and reliability of your study’s findings.
Author Response
Dear Editor and Reviewers:
First of all, we would like to thank you for reviewing the manuscript. We thank you for the time taken to read the document. Your comments are very instructive and inspired us a lot to work better. We have revised our manuscript after carefully reading your comments, and all the changes have been highlighted with different colors. Our response to each comment can be found in the following list of items.
We hope that the revision will improve the paper and the changes made in attempts to address your points are satisfactory. We would be glad to respond to any further questions and comments that you may have.
Response to the Comments of Reviewer # 2
Comment 1: Reference Accuracy (Lines 73-75): The paper by Zeng et al. [11], referenced in your manuscript, does not appear to accurately reflect the content it is cited to support. The cited study focuses on "grasp-and-place" using classical image matching for tactile grasping and reliable object grasping from clutter without precise placement requirements. However, this description better aligns with the following reference:
- Zeng et al., "Robotic Pick-and-Place of Novel Objects in Clutter with Multi-Affordance Grasping and Cross-Domain Image Matching," 2018 IEEE International Conference on Robotics and Automation (ICRA), Brisbane, QLD, Australia, 2018, pp. 3750-3757, doi: 10.1109/ICRA.2018.8461044.
Please review the reference and ensure that it corresponds accurately to the claims and context in which it is used.
Response: Thank you very much for your insightful comment. We have revised the introduction of the reference Zeng et al. [11]. The paper did not consider the impact of sparse rewards on grasping operations. Our architecture is inspired by Zeng et al.’s study, we have reshaped the rewards on this basis. Please see Page 2, Red color, Section 2.
Comment 2: Clarification on Comparative Analysis (Line 110): The manuscript states, "More closely related to our work is that of Zeng et al. [11]." This implies a benchmark comparison with the method described by Zeng et al. However, it is unclear from the text whether this comparison is theoretical or involves actual training and testing under similar conditions. Additionally, it is not evident why the approach by Zeng et al. was not included as a benchmark in Table 4 alongside the other benchmarks ([29], [30], and [31]). For clarity and completeness, please specify whether Zeng et al.'s method was directly compared in your experiments, discuss the scenarios under which both methods were evaluated, and explain the rationale for including or omitting this work in the benchmark comparison presented in Table 4.
Response: Thank you very much for your insightful comment. Due to the editor's requirement for the article to compare research from recent years, the research work of Zeng et al. (2018) was not compared at the beginning. For clarity and completeness, the benchmark of Zeng et al.'s method was directly compared in our experiments. Please see Page 14 and Page 17, Red color, Section 5.3 and 5.5.
Comment 3: Inclusion of Benchmark Studies in Related Work: The benchmarks presented in the results (Table 4 for benchmarks [29], [30], and [31]) are not mentioned in the Related Work section. It is crucial for consistency and reader understanding that these benchmarks be discussed in the Related Work section if they are pivotal to the study’s evaluation.
Response: Thank you very much for your insightful comment. For clarity and completeness,we have interpreted the benchmarks [29], [30], and [31] in the Related Work section to facilitate readers' reading and understanding. Please see Page 2-3, Red color, Section 2.
Comment4: Presentation of Training and Testing Results: The results for benchmarks [29], [30], and [31] only disclose testing outcomes. For a comprehensive evaluation, it is advisable to include both training and testing results. Presenting the training environment and methodology would provide a more reliable basis for comparing the efficacy of the different approaches tested.
Response: Thank you very much for your insightful comment. We have added the comparison of training for novel unknown objects with benchmarks in simulation. Please see Page 14, Red color, Section 5.3.

Round 3
Reviewer 2 Report
Comments and Suggestions for Authors
No more further comments